# Detailed Structure and Pathophysiological Roles of the IgA-Albumin Complex in Multiple Myeloma

**DOI:** 10.3390/ijms22041766

**Published:** 2021-02-10

**Authors:** Yuki Kawata, Hisashi Hirano, Ren Takahashi, Yukari Miyano, Ayuko Kimura, Natsumi Sato, Yukio Morita, Hirokazu Kimura, Kiyotaka Fujita

**Affiliations:** 1Department of Health Sciences, Gunma Paz University Graduate School of Health Sciences, 1-7-1, Tonyamachi, Takasaki-shi, Gunma 370-0006, Japan; kawata@paz.ac.jp (Y.K.); hirano@yokohama-cu.ac.jp (H.H.); re-takahashi@paz.ac.jp (R.T.); miyano@paz.ac.jp (Y.M.); ay-kimura@paz.ac.jp (A.K.); 723.sato@gmail.com (N.S.); fujita@paz.ac.jp (K.F.); 2Laboratory of Public Health II, Azabu University School of Veterinary Medicine, 1-17-71, Fuchinobe, Chuo-ku, Sagamihara, Kanagawa 252-5201, Japan; y-morita@azabu-u.ac.jp

**Keywords:** multiple myeloma, IgA-albumin complex, mass spectrometry, docking simulation, oxidized albumin

## Abstract

Immunoglobulin A (IgA)-albumin complexes may be associated with pathophysiology of multiple myeloma, although the etiology is not clear. Detailed structural analyses of these protein–protein complexes may contribute to our understanding of the pathophysiology of this disease. We analyzed the structure of the IgA-albumin complex using various electrophoresis, mass spectrometry, and in silico techniques. The data based on the electrophoresis and mass spectrometry showed that IgA in the sera of patients was dimeric, linked via the J chain. Only dimeric IgA can bind to albumin molecules leading to IgA-albumin complexes, although both monomeric and dimeric forms of IgA were present in the sera. Molecular interaction analyses in silico implied that dimeric IgA and albumin interacted not only via disulfide bond formation, but also via noncovalent bonds. Disulfide bonds were predicted between Cys34 of albumin and Cys311 of IgA, resulting in an oxidized form of albumin. Furthermore, complex formation prolongs the half-life of IgA molecules in the IgA-albumin complex, leading to excessive glycation of IgA molecules and affects the accumulation of IgA in serum. These findings may demonstrate why complications such as hyperviscosity syndrome occur more often in patients with IgA dimer producing multiple myeloma.

## 1. Introduction

Multiple myeloma is characterized by production of abnormal and clonal immunoglobulins, including Immunoglobulin A (IgA) and Immunoglobulin G (IgG) [1]. These abnormal proteins may result in complications such as amyloidosis, nephropathy, and hyperviscosity syndrome [2,3]. Indeed, IgA produced in multiple myeloma patients has been determined to be responsible for these complications [4].

IgA is classified into two types, monomeric and dimeric [5]. Most monomeric IgA is produced by non-mucosal lymphoid tissue and is distributed in various body fluids such as serum. Meanwhile, dimeric IgA is secreted by mucosal tissues, such as the gut, mammary glands, and nasopharyngeal/oral tissues [5]. The IgA dimers bind each other at the Fc region through a protein called the J chain [5]. Dimeric IgA plays an important role in mucosal immunity against pathogens, such as bacteria and viruses. IgA can also bind to various proteins, resulting in IgA-protein complexes [6]. Indeed, previous reports have shown that IgA can bind to albumin, α-lipoprotein, haptoglobin, anti-hemophilia protein, and α1-glycoprotein [6]. However, the function of these IgA-protein complexes is not clear [6].

Albumin is a common protein and broadly distributed in bodily fluids [7]. This protein acts as a carrier protein and can bind to various drugs, hormones, free fatty acids, and metal ions [8]. Moreover, previous reports have demonstrated that the oxidized form of albumin is associated with the pathophysiology of various diseases, including nephropathy and dyslipidemia [9]. Notably, albumin may be oxidized at specific amino acid residues, such as Cys34 [9]. However, it is not known whether oxidized albumin is associated with the pathophysiology of multiple myeloma or its complications. Moreover, the half-life of albumin in the serum (approximately 20 days) is significantly longer than that of IgA (approximately 6 days). Moreover, macromolecules such as IgA can alter the viscosity of body fluids, including serum [10,11]. Thus, IgA-albumin complexes may accumulate in body fluids during multiple myeloma, leading to complications such as hyperviscosity syndrome. Excessive glycation of the proteins including immunoglobulin may alter their functions [12]. However, pathophysiological roles of the glycated immunoglobulin in the patients with multiple myeloma is not exactly known.

In general, protein structures are closely linked to their functions. Therefore, in this study, to better understand the pathophysiology of multiple myeloma and hyperviscosity syndrome, we performed a detailed structural analysis of the IgA-albumin complex produced in multiple myeloma patients using proteomics and bioinformatics technologies.

## 2. Results

### 2.1. Confirmation of Albumin, IgA, and IgA-Albumin Complex in the Serum of Patients with Multiple Myeloma

To confirm the presence of albumin, IgA, and IgA-albumin complexes in sera, we performed immunofixation electrophoresis using polyclonal anti-albumin and anti-human IgA antibodies (Figure 1A–D). Using an anti-human IgA (α chain) antibody, we observed a single band corresponding to IgA in the sera, while using anti-human albumin antibody, double bands of IgA-albumin complex and albumin were observed (Figure 1C). Moreover, after the purification of IgA-albumin complexes from the sera, a single band corresponding to the IgA-albumin complex was observed (Figure 1D,E). Similar data were obtained in all four samples. On the other hand, no detection of IgA-albumin complex was observed in serum from healthy patients (data not shown). These results suggest that the serum collected from patients with multiple myeloma contained IgA-albumin complexes.

### 2.2. Western Blotting Analysis of IgA-Albumin Complex

To estimate molecular weight of the IgA-albumin complexes in sera, we performed SDS-PAGE electrophoresis and subsequent Western blotting (Figure 2A,B). A single band was observed at approximately 370 kDa in the purified samples of the IgA-albumin complexes from the sera of multiple myeloma patients. This value was approximately compatible with the molecular weight of the dimeric IgA (320 kDa) plus human serum albumin (66 kDa). Moreover, to reduce these complexes, the purified samples were treated with 2-mercaptoethanol (2-ME). As a result, we observed bands at around 65 kDa or 55 kDa (the size of the α-chain of IgA) (Figure 2C,D). These results implied that the IgA in these complexes was dimeric, and that the IgA-albumin complex contained disulfide bonds.

### 2.3. Comprehensive Analyses of the Proteins in Sera Using Mass Spectrometry

We analyzed proteins found in the sera of patients with multiple myeloma and normal adults using mass spectrometry (Table 1 and Appendix A). When purified IgA-albumin complexes cut from the gels after SDS-PAGE were digested and analyzed by mass spectrometry, the J chain (P01591) was detected in addition to IgA α chain (P01876), κ chain (P01834), and albumin (P02768). When monoclonal IgA purified from the same patient’s serum was digested in solution and analyzed by mass spectrometry, IgA α chain and κ chains were detected, but albumin and the J chain were not. These results indicate that only dimer-type IgA with the J chain forms IgA-albumin complexes. In addition, when IgA and albumin were purified from the sera of healthy individuals, in which monomer-type IgA comprised most of the serum, no albumin or J chain proteins were detected in the digested IgA by mass spectrometry, and no IgA was detected in the digested albumin. Moreover, α1-antitrypsin (P01009), α2-macroglobulin (P01023), and complement C3 (P01024) are present in large amounts in serum, and are likely to be detected non-specifically during purification because they are large molecular weight proteins that increase with inflammation.

### 2.4. Docking Simulation Analyses and Molecular Modeling of IgA-Albumin Complex

To analyze molecular interactions between IgA and albumin, we performed in silico docking simulations. In the resultant complex structure, the Cys311 residue of IgA was in close proximity to the Cys34 residue of albumin with the distance between the two sulfur atoms being 9.8 Å (Figure 3a). Many noncovalent interactions between the albumin and J chain were also observed (Figure 3b), and the binding affinity was estimated as −9.5 kcal/mol. These observations strongly suggest that the IgA molecules dimerized via J chain can form a stable complex with an albumin molecule.

### 2.5. Fructosamine Levels in the Sera of Patients with Multiple Myeloma Characterized by Abnormal IgA Production

Significantly elevated fructosamine levels in the serum of all patients were observed (467 ± 25 μM, reference values; 245 ± 19 μM). These results suggested that glycation of proteins in the sera was increased.

## 3. Discussion

In this study, we performed detailed molecular analyses of the IgA-albumin complex contained in the sera of the patients with multiple myeloma. First, IgA in the sera from these patients was revealed to be dimeric in nature, bound by the J chain protein. Second, although both monomeric and dimeric forms of IgA were found in the sera, only dimeric IgA could bind to albumin molecules to form IgA-albumin complexes. Third, simulated molecular interaction analyses suggested that these interactions between dimeric IgA and albumin occurred via not only disulfide but also noncovalent bonds (Figure 3). Additionally, specific cysteine residues, Cys311 of IgA and Cys34 of albumin, were predicted to form the disulfide bond, leading to an oxidized form of albumin. Oxidized albumin has been associated with the pathophysiology of complications of multiple myeloma, although the etiology of these diseases is not fully understood [13]. Moreover, significantly elevated fructosamine levels were observed in the sera of these patients, suggesting that IgA and accumulated protein-protein complexes may have increased half-lives in the sera. Glycation of IgA caused by prolongation of the half-life may also contribute to increased fructosamine levels. Finally, increased levels of IgA-albumin complexes due to prolonged half-life may promote hyperviscosity syndrome, a major complication in patients with multiple myeloma characterized by abnormal IgA production. To the best of our knowledge, this is the first study suggesting that dimeric IgA in multiple myeloma may associate with the complication of hyperviscosity syndrome.

Monomeric IgA is an immunoglobulin approximately 150 kDa in size. When dimerized via the J chain, the resulting IgA structure is around 300 kDa, and these molecules become resistant to proteolytic enzymes such as trypsin and airway mucosal serin protease (i.e., Transmembrane protease serin2 [TMPRSS2]) [14]. Dimeric IgA is secreted by various mucosal tissues, including in the gastrointestinal and respiratory tracts, and may play an important role in mucosal immunity against various pathogens [15]. Increased dimeric IgA in the serum may be associated with various complications of multiple myeloma [16]. In this study, we observed that dimeric IgA could bind to albumin, leading to the formation of IgA-albumin complexes. Moreover, our simulations predicted detailed molecular interactions between IgA and albumin in this complex.

Albumin is a common major protein distributed in bodily fluids, including sera. Albumin protein can bind various materials, including drugs, bilirubin, metal ions, and other proteins, and acts as a functional transporter [17]. Previous reports have shown that albumin has three binding sites (site I, II, and III) [18,19]. Moreover, an amino acid residue, Cys34, can bind to the SH group of ligands or to nitric oxide [9]. In the present study, we found that albumin could bind to dimeric IgA in the sera of patients with multiple myeloma. Moreover, we found that dimeric IgA is able to form stable complex with albumin by the docking simulation. These results are partly compatible with an earlier report [20]; however, that report did not identify the binding site between IgA and albumin [19]. Previous reports have also suggested that albumin oxidized at the Cys34 residue may be associated with diseases such as nephropathy and dyslipidemia [9]. The formation of a disulfide bond between IgA and albumin leads to oxidized albumin [21]. Therefore, the IgA-oxidized albumin complex may be associated with the pathophysiology of complications in patients with multiple myeloma characterized by abnormal IgA production, although the physiological roles of oxidized albumin have not been elucidated [21].

Then what interactions are responsible for the formation of IgA-albumin complex? The distance (9.8 Å) obtained from the HADDOCK analysis was the one between 2 sulfur atoms of Cys34 of albumin and Cys311 of IgA. The binding energy reported by HADDOCK software refers to the sum of all interactions between the atoms of albumin and the atoms of IgA, but not the particular binding energy between Cys34 of albumin and Cys311 of IgA. Indeed, as shown in Figure 3b and Appendix A, many intermolecular interactions were observed between albumin and the J chain of IgA. Moreover, as shown in Figure 2, the IgA-albumin complex dissociated by the addition of 2-mercaptoethanol, strongly suggesting that the binding of the complex was of covalent (disulfide) nature. The dissociation constant between albumin and IgA can be calculated as 1.08 × 10^−7^ M from the binding energy, −9.5 kcal/mol, reported by the HADDOCK software. This value suggests that the binding is strong enough to form a stable IgA-albumin complex. We thus infer that the internal motion of the complex enables the mutual approach of the 2 sulfur atoms at 9.8 Å distance in the docked structure to form an -S-S- bond (2.05 Å). We do not expect electrostatic attraction between the 2 sulfur atoms, because the sulfur atom in a cysteine residue is always charged negative (−1 in the deprotonated state and −0.57 in the protonated state) [22].

The half-life of serum albumin is estimated to be around 20 days, while the half-life of IgA is around 6 days [10,11]. Therefore, when IgA molecules bind to albumin, the half-life of the IgA may be prolonged, leading to excessive glycation [23]. Moreover, a previous report suggested that fructosamine levels in sera were significantly elevated in multiple myeloma patients with abnormal IgA production [24]. Indeed, fructosamine levels were found to be significantly elevated in sera in the present study. Hyperviscosity syndrome is a common complication in patients with multiple myeloma [25]. This complication may correlate with disease survival [26] because of the prolonged half-life albumin-bound IgA. Previous reports suggested that hyperviscosity syndrome in multiple myeloma may be responsible for increased blood cells (erythrocytes, leukocytes, and platelets) and macromolecules (mainly immunoglobulins) in the blood [25]. Previous reports also suggested that blood hyperviscosity may be partly responsible for increased IgA-protein complexes in patients with multiple myeloma [27]. These findings can be compatible with our results. Moreover, previous reports showed that increased IgA levels in the sera of patients with multiple myeloma are associated with hyperviscosity syndrome [28]. In these reports, IgA levels of patients with greater than 4000 mg/dL [28] were similar to those in our data from patients with greater than 4200 mg/dL. To actually measure blood viscosity, relatively large amounts of blood samples are needed (around 7 mL) [29]; unfortunately, one limitation of the present study is that we did not have adequate sample volume to measure serum viscosity.

In conclusion, we present here a hypothesis as to why complications such as hyperviscosity syndrome frequently accompany dimeric IgA production in multiple myeloma. Our results and speculations may contribute to the understanding of the pathophysiology of multiple myeloma and disease prognosis.

## 4. Materials and Methods

### 4.1. Subjects

Sera were obtained from four patients ranging in age from 52 to 84 (69.8 ± 13.7 years) with multiple myeloma characterized by abnormal IgA production. All four sera from patients with high IgA (greater than 4000 mg/dL). These patients were not clinically diabetic and had blood sugar of 98.5 ± 35.4 mg/dL. Normal sera were also collected from 10 adults ranging in age from 18 to 35 (25.3 ± 6.9 years). Informed consent was obtained from all participants, which was obtained from the subjects or their legal representatives upon sample donation. All patient data were anonymized. Because of the lack of written informed consent, the present study protocols were deliberated by the Ethics Committee on Human Medical Research of Gunma Paz University (Gunma, Japan), which ruled that this study did not infringe upon the patient’s rights, and was approved by the research ethics committee (approval no. PAZ-18-9). All methods were performed in accordance with the approved guidelines.

### 4.2. Confirmation of IgA-Albumin Complexes in the Sera of Patients with Multiple Myeloma by Immunofixation Electrophoresis

Serum was diluted 21-fold with phosphate-buffered saline (PBS). Electrophoresis was performed in 0.06M barbital buffer (pH 8.6) using a commercially available kit (Quick gel IFE kit, HERENA Laboratory, Urawa, Japan). Serum (3 μL) was applied and electrophoresed at a constant voltage of 90 V for 20 min. After electrophoresis, the gel was incubated with anti-human IgA (α chain) and anti-human albumin antibodies for 10 min. After reaction, the gel was stained with acid blue dye for 2 min and decolorized with 10% acetic acid to minimize background staining.

### 4.3. Purification of IgA, IgA-Albumin Complex, and Free Albumin from Serum

Purification steps for IgA, the IgA-albumin complex, and free albumin are shown in Figure 4. First, proteins were concentrated from serum by affinity chromatography using Cibacron Blue F3G-A columns (Amersham Biosciences, Little Chalfont, Britain) (Figure 4a). The column washed with 0.05 M Tris-HCl buffer (pH 7.0) containing 0.1 M potassium chloride, and the bound proteins (albumin, monoclonal IgA, and IgA-albumin complex) were eluted by 0.05 M Tris-HCl buffer (pH 7.0) containing 1.5 M potassium chloride. Proteins were concentrated with an Amicon Minicon-B15 concentrator (Amicon Division, Beverly, MA), and then purified with Jacalin-Agarose (Funakoshi Corporation, Tokyo, Japan) (Figure 4b). The unbound protein (free albumin) was eluted with 1M phosphate buffer (pH 7.2), and the bound proteins (IgA-albumin complex and monoclonal IgA) were eluted with 0.1 M phosphate buffer (pH 7.2) containing 0.8 M galactose. Finally, the IgA-albumin complex and the monoclonal IgA were separated using an anti-human albumin antibody affinity column (Affi-Gel Hz Hydrazide Gel (BIO-RAD, California, America)) (Figure 4c). Unbound protein (monoclonal IgA) was eluted by 0.02M Tris-HCl buffer (pH 7.5) containing 0.005% Briji-35, and the bound protein (IgA-albumin complex) was eluted with a 0.1 M glycine hydrochloride buffer (pH 2.8).

### 4.4. SDS-PAGE and SDS-Agarose-Polyacrylamide Hybrid Gel Electrophoresis Analysis

A total of 10 μg of purified IgA-albumin was mixed with sample buffer (0.375 M Tris-HCl buffer (pH 8.8), 5% 2-Mercaptoethanol, 2% SDS, 20% glycerol). SDS-polyacrylamide gel electrophoresis (PAGE) was performed using 5–20% polyacrylamide gradient gels. SDS-agarose-polyacrylamide hybrid gels containing 5 M urea were prepared by mixing 4% acrylamide and 1% agarose for separation of high-molecular-weight proteins [30]. The purified samples were diluted with 0.5 M Tris-HCl buffer (pH 6.8) and then mixed with sample buffer without reducing agent (4% SDS, 20% glycerol, 0.02% bromophenol blue in 0.1 M Tris-HCl buffer (pH 6.8)). Each diluted sample was heat-denatured at 95 °C for 5 min, and electrophoresis was performed at a constant current of 20 mA for 2 h.

### 4.5. Western Blotting

Proteins separated by SDS-PAGE were transferred to polyvinylidene difluoride (PVDF) membranes (MERCK, Tokyo, Japan) for 90 min at 200 mA constant current using a Trans-Blot SD cell (BIO-RAD, California, America). Immunostaining was performed using rabbit anti-human polyclonal antiserum (MBL, Nagoya, Japan), for primary immunoreaction, and Horseradish peroxidase (HRP)-labeled goat anti-rabbit IgG (H + L) serum (MBL) for secondary immunoreaction. After washing the membranes with PBS-T (pH 7.4), bound antibodies were detected with POD Immunostain Set (FUJIFILM Wako, Osaka, Japan).

### 4.6. In-Gel Digestion

Gel lanes were cut into 1 mm square gel pieces. Gel pieces were decolorized with 50 mM NH_4_HCO_3_/60% acetonitrile (ACN) at 37 °C for 30 min four times and lyophilized using a vacuum freeze-dryer (Techcorp FD-3-85-MP). Proteins were digested with trypsin solution (25 ng/μL) at 37 °C for 16 h. Peptide digests were desalted and concentrated using C18 stage tips prior to LC-MS/MS analysis [31]

### 4.7. Liquid Chromatography/Tandem Mass Spectrometry (LC-MS/MS)

LC-MS/MS analysis was performed as reported previously [32]. Digested peptides were separated by a 115-min acetonitrile gradient from 4% to 40% using an Ultimate 3000 nano-HPLC system (Dionex Softron, Germany) with a Nanoscale C18 capillary LC column (Acclain PepMap 100 C18 column, 75 µm id, 150 mm length, 3 µm particle size, 100 Å). The liquid was directly electrosprayed with electrospray ionization source in positive ion mode using a SilicaTip PicoTip nanospray emitter (10 µm id, top non-coated; New Objective Inc., Woburn, MA, USA) and analyzed on an LTQ Orbitrap Velos (Thermo Fisher Scientific, Bremen, Germany). MS/MS analysis was performed in data-dependent scanning mode with a full-range scan (m/z range from 350 to 1200), followed by product ion scans for the 15 most intense ions selected from the MS-scan spectra. Peptides were identified using MaxQuant ver.1.6.2.10 [33]. Peptides were queried against the Uniprot human database using default settings with slight modifications as follows: main search peptide tolerance of 6 ppm (http://www.uniprot.org/). Finally, peptides with fewer than two missed trypsin cleavages, a false discovery rate < 0.01, and an Andromeda score > 100 were selected for analysis. Proteins identified by only one peptide were discarded, and matches with at least two unique peptides (Andromeda score ≥ 100) were considered to be accurately identified [34].

### 4.8. Docking Simulation

Molecular docking simulation was used to identify the predicted binding site of the monoclonal IgA-albumin complex. The Fc regions and J chain from dimeric IgA (2QTJ) [35] and albumin (1AO6) [36] in the open configuration were downloaded from the protein data bank. The solvated docking software HADDOCK [37] was used to dock albumin with IgA in the presence of the J chain component. After docking simulation, PRODIGY software was used to predict the binding affinity for IgA and albumin. The average docking scores and residues predicted to take part in the interaction were also determined.

### 4.9. Measurement of Fructosamine in the Serum

Serum fructosamine levels in four patients with multiple myeloma characterized by abnormal IgA production were measured with Spotchem EZ SP-4430 (Arkray, Tokyo, Japan) as previously described [38].

## Figures and Tables

**Figure 1 ijms-22-01766-f001:**
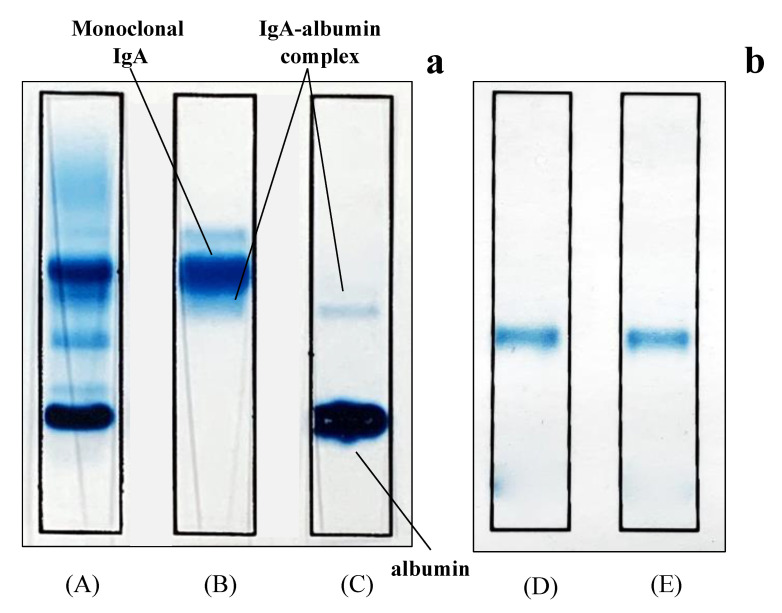
Immunofixation electrophoresis of patient sera (**a**) and of purified IgA-albumin complex (**b**). Proteins were analyzed by immunofixation electrophoresis using acid blue stain (lane **A**), anti-IgA (α chain) (lanes **B**,**D**), and anti-albumin (lanes **C**,**E**) antiserum.

**Figure 2 ijms-22-01766-f002:**
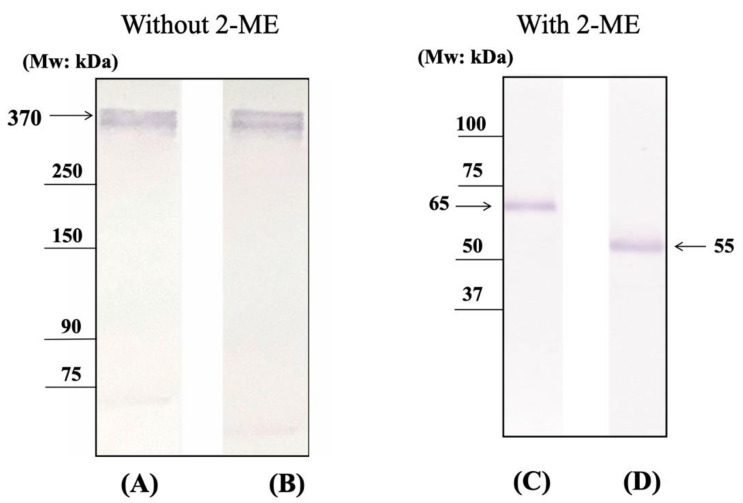
Western blot analysis of the purified IgA-albumin complex. IgA-albumin complexes were analyzed by SDS-PAGE and immunoblotted using anti-albumin (lanes **A**,**C**) and anti-IgA (α chain) (lanes **B**,**D**) antiserum.

**Figure 3 ijms-22-01766-f003:**
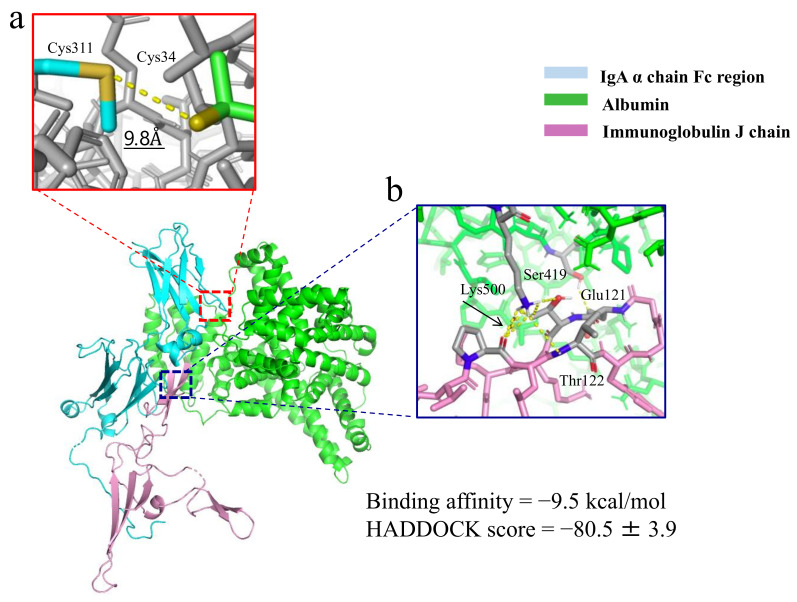
Predicted structure of the IgA-albumin complex (colored cartoon). In the enlarged panel, the Cys311 residue of IgA was in close proximity to the Cys34 residue of albumin (**a**). The other panel suggests that there are many noncovalent bonds between the J chain and albumin (**b**). An arrow indicates noncovalent bond.

**Figure 4 ijms-22-01766-f004:**
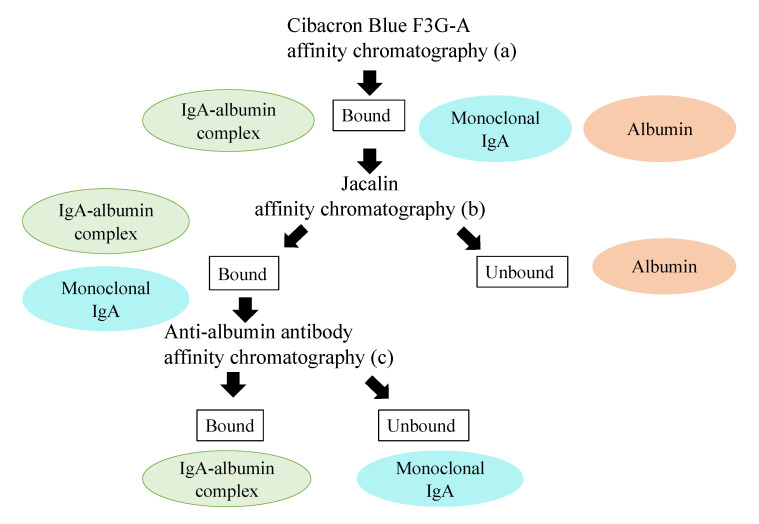
Procedure for purification of each protein from serum. Procedures are represented as (**a**–**c**).

**Table 1 ijms-22-01766-t001:** Proteins identified by mass spectrometry from each purification step (IgA-albumin complex, monoclonal IgA, albumin and polyclonal IgA).

Analyzed Sample Names	Protein IDs	Identified Protein Names	Gene Names	Unique Peptides	Mol. Weight * [kDa]
IgA-albumin complex from Patients sera	P02768	Serum albumin	ALB	21	69.37
	P01876	Ig alpha-1 chain C region	IGHA1	7	37.65
	P01834	Ig kappa chain C region	IGKC	7	11.77
	P01591	Immunoglobulin J chain	IGJ	7	18.10
	P01009	Alpha-1-antitrypsin	SERPINA1	8	46.74
Monoclonal IgA (albumin no-bound)from Patients sera	P01876	Ig alpha-1 chain C region	IGHA1	11	37.65
	P01834	Ig kappa chain C region	IGKC	7	11.77
	P01024	Complement C3	C3	7	187.15
	P01023	Alpha-2-macroglobulin	A2M	5	163.29
	P01009	Alpha-1-antitrypsin	SERPINA1	4	46.74
	P02647	Apolipoprotein A-I	APOA1	4	30.78
Albumin from Normal sera	P02768	Serum albumin	ALB	45	69.37
	P01023	Alpha-2-macroglobulin	A2M	8	163.29
	P02787	Serotransferrin	TF	12	77.06
	P10909	Clusterin	CLU	6	52.49
	P00738	Haptoglobin	HP	6	45.21
Polyclonal IgA from Normal sera	P01876	Ig alpha-1 chain C region	IGHA1	9	37.65
	P01834	Ig kappa chain C region	IGKC	4	11.77
	P01023	Alpha-2-macroglobulin	A2M	8	163.29
	P01009	Alpha-1-antitrypsin	SERPINA1	7	46.74

Note: Proteins identified by only one peptide were discarded, and only proteins that matched at least two unique peptides (Andromeda score ≥ 100) were included. Detailed peptide data are shown in the supplement (Appendix A). * The molecular weights as reported in the Uniprot database.

## Data Availability

The protein structural data in Figure 3 are available online at https://www.rcsb.org/structure/2QTJ and https://www.rcsb.org/structure/1AO6.

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
