# Peer review of "Detailed Structure and Pathophysiological Roles of the IgA-Albumin Complex in Multiple Myeloma"

_ijms, 2021, doi:10.3390/ijms22041766_

Round 1

Reviewer 1 Report

Major issues:
The authors sought to demonstrate the mechanisms of hyperviscosity syndrome (HVS) in patients with IgA multiple myeloma. The authors ascribed HVS to the formation of IgA-albumin complex in the serum of patients with multiple myeloma. Nevertheless, HSV is caused by an increase in cells (red blood cells, white blood cells, and platelet) or the protein content of the circulation. This increase thickens the blood and makes the patients more prone to HSV.

The authors concluded that the prolonged half-life of IgA due to albumin binding may have contributed to an increased incidence of HVS in IgA multiple myeloma. To prove this hypothesis, it is necessary to measure the degree of viscosity in the IgA-albumin complex and free monoclonal IgA in patients with myeloma.

In addition, the molecular weights of IgA-albumin complexes are exactly the same as the monoclonal IgA from the same myeloma patient (P01876, P01834 and P01009) despite the addition of albumin to IgA. Some of the free monoclonal IgA proteins are much heavier than IgA-albumin complexes (P01024, P01023). These discrepancies in the molecular weights should be accounted for in terms of risk factors associated with HVS occurrence.

Minor issues:
Figures are not properly cited in the manuscript.
Typos, punctuation errors, and spacing errors should be corrected accordingly.

Reviewer 2 Report

The authors study the IgA-albumin complex's role in the pathophysiology of multiple myeloma using experimental and computational investigative techniques.

Because of the topic covered and the methodologies used, the manuscript is written concisely. For example, mass spectrometry and computational investigations should be reported with more detail and in a more accurate manner.

Row 107-112, The authors should explain this concept more thoroughly.

Table 1 should be explained in more detail.

Chapter 2.4. This chapter should be reconsidered entirely, and more detail is needed. The authors should also explain how they can demonstrate a covalent bond between two cysteines from docking simulations. The energy value is so low (9.5 kcal/mole). This issue should also be corrected in the discussion section.

Row 119-202, a phrase not clear.  

Round 2

Reviewer 2 Report

The discussion of the interaction between cysteines 311 and 34 is still unclear. How do the authors calculate the energy? Which atoms are involved in the bonding/interaction? Does an -S-S- bond form or is it just an electrostatic interaction. If it is a sulfur-sulfur electrostatic interaction, what is the difference in electronegativity between the atoms involved in the interaction? The authors think that interaction at the distance of 9.8 Angstrom can have an 8.8 kcal/mole energy. Let us compare this value with a hydrogen bridge interaction at a distance of 3.5 Angstrom. We find a value in the range of 5-8 kcal/mole, and it depends on the electronegativity of the atoms involved.
